# Tide-Inspired Path Planning Algorithm for Autonomous Vehicles

**Heba Kurdi [1,2,\*]**, **Shaden Almuhalhel [1]**, **Hebah Elgibreen [2,3,4]**, **Hajar Qahmash [1]**, **Bayan Albatati [1]**, **Lubna Al-Salem [1]** and **Ghada Almoaiqel [1]**

1   Computer Science Department, College of Computer and Information Sciences, King Saud University, Riyadh 11451, Saudi Arabia; 435201535@student.ksu.edu.sa (S.A.); Qahmash.Hajar@Gmail.com (H.Q.); albatati.bayan@gmail.com (B.A.); binsalem.lubna@gmail.com (L.A.-S.); galmoaiqel@gmail.com (G.A.)
2   Mechanical Engineering Department, Massachusetts Institute of Technology (MIT), Cambridge, MA 02139, USA; hjibreen@ksu.edu.sa
3   Information Technology Department, College of Computer and Information Sciences, King Saud University, Riyadh 11451, Saudi Arabia
4   Artificial Intelligence Center of Advanced Studies (Thakaa), King Saud University, Riyadh 11451, Saudi Arabia
\*   Correspondence: hakurdi@mit.edu

**Abstract:** With the extensive developments in autonomous vehicles (AV) and the increase of interest in artificial intelligence (AI), path planning is becoming a focal area of research. However, path planning is an NP-hard problem and its execution time and complexity are major concerns when searching for optimal solutions. Thus, the optimal trade-off between the shortest path and computing resources must be found. This paper introduces a path planning algorithm, tide path planning (TPP), which is inspired by the natural tide phenomenon. The idea of the gravitational attraction between the Earth and the Moon is adopted to avoid searching blocked routes and to find a shortest path. Benchmarking the performance of the proposed algorithm against rival path planning algorithms, such as A\*, breadth-first search (BFS), Dijkstra, and genetic algorithms (GA), revealed that the proposed TPP algorithm succeeded in finding a shortest path while visiting the least number of cells and showed the fastest execution time under different settings of environment size and obstacle ratios.

**Keywords:** path planning; nature-inspired algorithms; autonomous vehicles; transportation

## 1. Introduction

Autonomous vehicles are anticipated to be profoundly entrenched in Industry 4.0, which demonstrates the significant impact of efficient path planning. The escalating upsurge in the environmental complexity and the rapid evolution of application areas, such as transportation, agriculture, manufacturing, and patrolling, has magnified the intricacy of the path planning problem significantly. In the path planning problem, an agent should autonomously find the shortest path from a start point (origin) to a goal point (destination), while avoiding obstacles between these points at a minimum cost [1].

Path planning is an NP-hard problem [2], as the execution time is a major concern when trying to find an optimal solution [3]. Thus, a plethora of path planning algorithms have been proposed. Based on the literature, path planning algorithms can be classified as exact methods, heuristic methods, and hybrid methods. Exact methods find the exact optimal path. They are simple but expensive and cannot handle uncertainty. Exact methods, such as A\*, breadth-first search (BFS), and Dijkstra, involve a trade-off between solution accuracy on one hand and time and energy on the other hand [4]. Thus, they are not applicable to uncertain or large environments. In contrast, heuristic methods, which introduce heuristics to relax the search and reduce the execution time, such as the genetic algorithm (GA), are more flexible. However, they cannot guarantee finding the optimal

solution that uses less time, space, and expanded nodes. Moreover, poorly designed heuristic methods can result in large computational overheads [5]. To combine the advantages of multiple methods, while reducing their drawbacks, hybrid methods were introduced. However, even though hybridization has shown better results, the complexity of hybrid algorithms is usually high and the compatibility of some of the algorithms is not validated yet [6]. Therefore, path planning is still an open research area, and more development is still needed to reduce consumed resources and execution time.

This paper introduces a path planning heuristic algorithm, tide path planning (TPP), inspired by the tide phenomenon in nature. In the tide phenomenon, water, primarily ocean water, moves in a certain direction under the effect of two main forces: the Moon's gravitational pull and centrifugal force. The main factor that determines what exerts the greater influence is the distance, i.e., water nearer to the Moon is pulled towards it, while water farther away is affected by the centrifugal force. In a similar manner, an agent (water) is pulled towards the goal, which represents the Moon. However, if the agent encounters an obstacle, which represents a centrifugal force, it is directed away from the goal. It is important to note that the Sun's gravitational force also influences tides; however, as the Moon has the greatest influence, we have focused on the Moon instead.

To extensively analyze the performance of the proposed algorithm, long-standing path planning exact and heuristic algorithms were used as benchmarks, including A*, breadth-first search (BFS), Dijkstra, and genetic algorithms (GA). Three different performance measures were collected, including the number of visited cells (as a measure of energy efficiency), path length (to test optimality), and algorithm execution time (to assess the algorithm complexity). Two main path planning algorithm design issues were considered, which include scalability to wide area sizes and robustness under a large number of obstacles. The experimental results show that TPP finds a shortest path, if it exists, at significantly less execution time and energy consumption, when compared with the benchmarks, under different settings of area size and obstacle ratios.

The remainder of this paper is structured as follows: Related literature is reviewed in Section 2. In Section 3, our proposed algorithm is introduced, while its evaluation framework and experimental results are presented in Sections 4 and 5, respectively. Finally, the paper is concluded, and future directions are outlined in Section 6.

## 2. Literature Review

Path planning algorithms can be classified as exact, heuristic, and hybrid methods. Exact methods find the optimal path without approximation [7]. On the other hand, heuristic methods include approximation and can find a near-optimal or a "good" solution [8]. Meanwhile, hybrid methods combine more than one method to benefit from their advantages and overcome their limitations. In [4], Russell and Norvig described different classical graph search algorithms for path planning. For example, BFS is a well-known exact search algorithm that expands nodes in a FIFO (first-in, first-out) manner. BFS is optimal when the cost is a non-decreasing function. However, memory requirements and execution time are significant concerns. The depth-first search (DFS) is an exact method that searches nodes in a LIFO (last-in, first-out) manner. In infinite state spaces, DFS fails when a non-goal path is encountered. The Dijkstra algorithm, which is also an exact method, uses a greedy approach to solve the shortest path problem in a graph. However, the Dijkstra algorithm searches blindly; thus, computation time increases significantly [9]. The A* search algorithm [10] extends the Dijkstra algorithm and uses a cost function to assess the node's quality in the process of path searching. The algorithm selects the node with minimum cost as the next node to expand. Then, it continues to search from the next node until the target point is reached. The A* search algorithm is both complete and optimal, in terms of time and space, on graphs that are locally finite, where the heuristics are admissible and monotonic. Additionally, the A* algorithm is faster than the other exact search algorithms [4]; however, its computational cost is a concern. Saicharan et al. [6] introduced another algorithm based on the A* algorithm and showed the impact of introducing a

parent node on the heuristic function. The experimental results demonstrated that the proposed algorithm solved the poor real-time capacity of the A* algorithm at the expense of path cost. In [11] the constrained D* algorithm for real time re-planning of optimal paths is presented. However, the simulation results suggest that the algorithm cannot handle multiple constraints.

Pragnavi et al. [7] indicated that, even though exact methods are simple, their execution time is expensive. Thus, heuristic methods are proposed which utilize practical techniques or shortcuts to produce a good solution. Specifically, nature-inspired heuristics are attracting attention in path planning. They mimic biological systems, chimerical interactions, or physics phenomena to generate reasonable solutions at a low cost [5,12]. A survey of the literature that focused on nature-inspired path planning heuristics is presented in [8], which found that three main approaches are dominant: genetic algorithms (GA), particle swarm optimization (PSO), and ant colony optimization (ACO). Typically, GAs involve a trade-off between execution time and solution quality; therefore, they converge to a premature solution because population diversity cannot be controlled. PSO is a key algorithm for the path planning optimization problem. However, PSO takes a long time to converge to a global optimal solution. Furthermore, designing an efficient PSO fitness function is challenging [6]. ACO suffers from extended execution time in large environments. Additionally, tuning its multiple parameters to fit the problem is exacting.

Although less frequently used, there exist some other nature-inspired heuristics. The plant growth path planning (PGPP) algorithm [13] was inspired by plant growth and the way in which plants seek any light source. The experimental results demonstrated that the PGPP algorithm could improve path planning results. The results suggested that this algorithm is more suitable for mission planning systems with stringent time limits, such as online path planning. Another path planning algorithm, based on chemotaxis, has been proposed in [14]. In nature, some slime molds use this chemical process to find the shortest path between their locations and food sources. The chemotaxis-based algorithm determines the direction of the robot based on the direction with the highest attraction. The experimental results suggested that the algorithm always finds the shortest path, despite obstacles. Purcaru et al. [15] proposed a gravitational search algorithm (GSA), a physics-based path planning algorithm inspired by gravity law. The experimental results showed that the path length of the GSA algorithm is comparable to that of PSO, but with less execution time. A cuckoo-inspired algorithm has also been proposed in [16]. This algorithm is inspired by the way cuckoos lay their eggs in the nests of other host birds. When tested, the proposed algorithm efficiently generated feasible solutions and guaranteed global convergence.

A hybrid path planning algorithm is proposed in [12] which combines the A* and D* lite [13] algorithms. The proposed hybrid algorithm requires less computation time and demonstrated better re-planning results compared to the original algorithms. An improved hybrid A* algorithm was proposed in [17]. When evaluated, the new algorithm showed improved performance by reducing the number of steering actions and the maximum curvature of the paths. In [18] a path planning algorithm that hybridizes the rapidly exploring random tree algorithm (RRTs) and the differential evolution (DE) algorithm was presented. The hybrid algorithm shows the capability to generate a fast and optimal 3D collision-free path under complex environments. A hybrid path planning algorithm, called ACO-GA, was proposed in [19] based on ACO and genetic algorithm (GA). ACO-GA outperforms ACO for a large number of ants, while its performance degrades when the number of ants is small. In [2], GA was integrated with membrane computing to solve the path planning problem. Intensive tests were conducted in complex environments to find that the proposed algorithm can find better paths compared to the original methods. However, there was a trade-off between the execution time and the quality of the solutions. An improved ant colony algorithm for robot path planning was proposed in [20]. It is a combination of the ACO algorithm and the geometry optimization method. The algorithm considers static obstacles and generates good solutions quickly, while lowering the risks

of trapping into a local optimum. A hybrid mobile robot path planning algorithm was proposed in [3] that combines two nature-inspired meta-heuristics, based on cuckoo and bat behaviors, i.e., the parasitic behavior of cuckoos and the echolocation behavior of bats. The algorithm was further improved and tested using a real robot in [21]. The test results indicated that, although accuracy has been improved, the two algorithms require tuning parameters that might affect their performance in a complex environment.

Badue et al. [22] stated that autonomous cars are among the most intelligent systems today and although a large body of research has contributed to the current state, still much has to be achieved to reach the industrial and academic visions for them. According to Contreras-Cruz et al. [23], more research needs to be conducted in path planning to improve local exploration processes and execution time. Thus, this field is still an open research area, and more development is still needed to find the optimal solution, while reducing consumed resources and improving the execution speed.

## 3. Algorithm Design

This paper proposes a static path search heuristic algorithm to find a path, from an origin to a destination, considering two dimensional (2D) maps. The proposed algorithm, TPP, is inspired by the tide phenomenon. Tides are the regular changes in water levels, most noticeably in large bodies of water, i.e., oceans. Such changes are primarily caused by the gravitational force of the Moon, which creates what is known as a tidal bulge, where water closer to the Moon is pulled towards it, while water farther away from the Moon is affected by the centrifugal force, produced at any point on the Earth, which directs them away from the Moon [24], as shown in Figure 1.

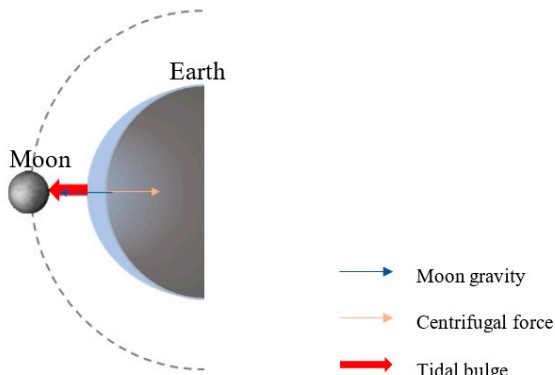

**Figure 1.** Tide phenomenon.

In a similar manner, in the proposed algorithm, the agent (water) is pulled to the goal (Moon), while obstacles (centrifugal force) direct the agent away from the goal. Here, the idea is to give cells around obstacles a comparatively high cost; thus, the agent will be repelled away from them, and cells leading to the goal will have lower cost; thus, the agent will be attracted to them. It is important to note that the Sun's gravitational force also influences tides, but to a lesser extent, and therefore it has not been considered in our analogy.

The environment (the world) is represented as a grid of discrete cells. The start and goal points, as well as obstacles, are generated randomly for each run. The TPP algorithm comprises three main components—obstacle neighbors incrementor, path finder, and cell value calculator—as shown in Figure 2. Each component implements part of the TPP algorithm, as shown in Figure 3.

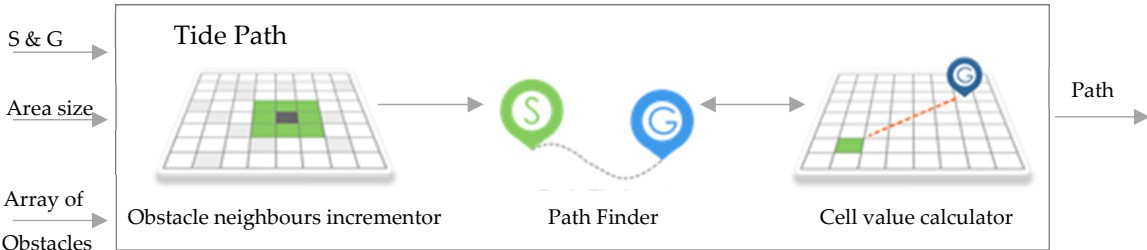

**Figure 2.** TPP algorithm components.

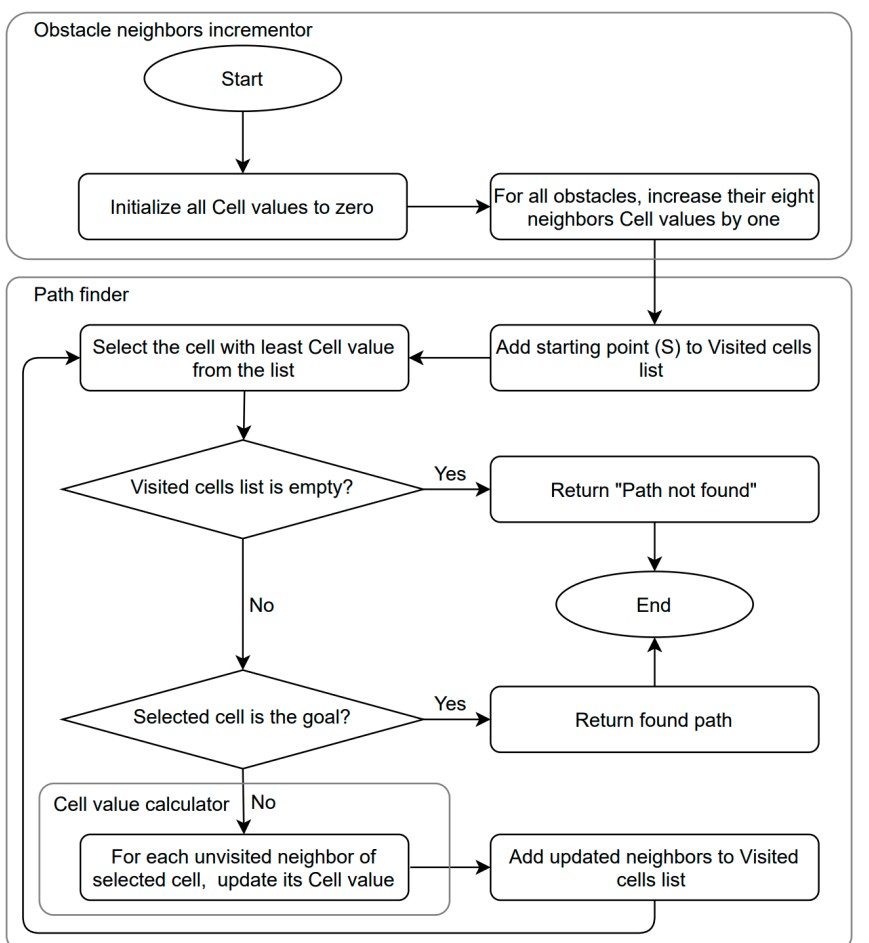

**Figure 3.** TPP algorithm flowchart.

Once the start, goal, and obstacles cells are determined, the obstacle neighbors incrementor increases the values of all immediate obstacle neighbor cells by one. Each obstacle has eight neighbors, i.e., the cells immediately on its left, right, up, down, and diagonal lines. Thus, the cost of the cells next to an obstacle will be higher than the other cells.

Then, the path finder searches for an optimal path from the start to the goal. It maintains a list of visited cells and starts by designating the starting point as the current cell. While the list is not empty, the visited cell with the minimum cost will be checked for whether it is the goal point. If this is true, then the sequence of cells visited from the starting point to the current point will be returned as the final path. Otherwise (i.e., if the currently selected cell is not the goal), then the cell value calculator will update the cell value of all unvisited neighbors of the current cell.

The cell value, $V(n)$, of each cell is calculated using the heuristic function in Equation (1).

$$\text{Cell value} = V(n) = D(n,g) + \left(1 - \frac{1}{D(n,g)}\right) \times \sqrt{V_{old}(n)} + C(s,n) \tag{1}$$

where $V_{old}(n)$ is the previous cell value, $D(n,g)$ is the Manhattan distance from cell $n(n_x,\ n_y)$ to the goal cell $g(g_x,\ g_y)$, which is calculated based on Equation (2).

$$D(n,g) = |n_x - g_x| + |n_y - g_y| \tag{2}$$

$C(s,n)$ is the cost to reach from the starting cell $s(s_x,\ s_y)$ to the neighbor cell, $n(n_x,\ n_y)$, calculated as the Manhattan distance between the two points, based on Equation (3).

$$C(s,n) = D(s,n) = |s_x - n_x| + |s_y - n_y| \tag{3}$$

After updating the cell values and computing the cost, the path finder removes the current "visited" cell from the visited cell list, adds updated neighbors to the visited cell list, marks the updated neighbors as visited, and continues to search for the goal by checking neighbors that minimize the total cost, until the path is found or no more cells are available to be visited.

## 4. Evaluation Methodology

The main design considerations in path planning algorithms are their scalability to large areas and their robustness under an extremely large number of obstacles. Therefore, different map sizes in the range $[2^3 \times 2^3.\ 2^9 \times 2^9]$ at different obstacle coverage ratios (10–40%) were investigated. Additionally, the performance of the proposed TPP algorithm was benchmarked against four well-established path planning algorithms—BFS, A*, Dijkstra, and GA algorithms—considering three performance measures:

- Number of visited cells: Counts the number of cells tested until the goal is reached during the algorithm's execution. This measure directly affects the number of CPU cycles and memory usage, which correlates with energy consumption (i.e., a lower number of cells indicates less energy consumption).
- Path length: Counts the number of cells in the final path between the goal and the starting point (a shorter path length indicates better results).
- Execution time: The execution time is the time from when the system calls the main function of an algorithm until the control returns to the caller. Thus, execution time indicates the algorithm speed (shorter time indicates a faster algorithm).

For each metric, scalability and robustness were evaluated, and the result distributions are illustrated using boxplots that also show the minimum, maximum, and median values of each algorithm.

## 5. Results and Discussion

In this section, the experimental results of the proposed TPP algorithm, compared with the benchmark algorithms, are presented based on three performance measures: the number of visited cells, the path length, and the execution time. It is important to note that, in all charts and graphs, a logarithmic scale is used for the y-axes due to the considerable variability in performance between the algorithms. All experiments were performed on an Intel Core i7 processor with 16 GB RAM. Each experiment was repeated several times and averages were calculated and visualized.

### 5.1. Number of Visited Cells

5.1.1. Scalability

This scalability test examines the performance of the algorithms, in terms of the number of visited cells, as the map size increases exponentially from $8 \times 8$ to $512 \times 512$. Figure 4a shows that, as expected, for all algorithms, increasing the map size increases the

path length, since more cells must be visited to reach the goal when the map gets larger. From the figure, it is clear that the TPP and A* algorithms yielded the best results in all map sizes. They showed nearly identical performance, with the TPP algorithm performing slightly better in large maps, i.e., maps larger than $32 \times 32$. This can be attributed to the centrifugal force property, where TPP could efficiently avoid obstacles in advance and reduce the time and energy required to discover blocked or longer paths. The Dijkstra and BFS algorithms also exhibit identical performance and come next after the TPP and A* algorithms. The performance gap is clearer in maps larger than $64 \times 64$, where the Dijkstra and BFS algorithms visit significantly more cells (increased from thousands to tens of thousands). For GA, the number of visited cells is significantly high, especially in large maps. This confirms the limitations of evolutionary-based algorithms in path planning, as discussed in [5]. In particular, in maps larger than $64 \times 64$, GA could not find a path and returned null values.

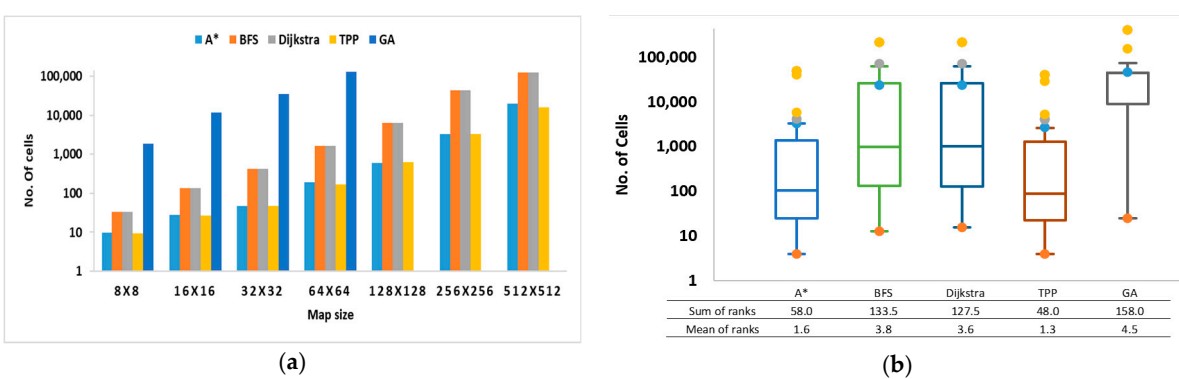

**Figure 4.** Scalability results: number of visited cells. (**a**) The average number of visited cells at different map sizes. (**b**) Distribution of results for the number of visited cells.

To gain more explicit understanding of the algorithms' behavior overall in the trials, the results distributions were visualized using the boxplots in Figure 4b. The figure shows that the number of visited cells in both the TPP and A* algorithms is less dispersed than in the BFS and Dijkstra algorithms. Additionally, the TPP and A* algorithms also have lower minimum and maximum values, which indicates that their numbers of visited cells are generally lower than the other algorithms. When comparing the TPP to the A* plots, they look similar in distribution and in minimum and maximum values; however, in the TPP plot, the median—represented by the horizontal line in the box—is not in the middle of the box, which indicates that the results are skewed to the smaller values. Hence, the number of visited cells tends to be lower in the TPP algorithm. GA displays the highest minimum and maximum values with the least distribution as, in all scenarios, the algorithm visited significantly more cells to find a shortest path or failed to find one.

### 5.1.2. Robustness Test

This robustness test observes the algorithms' performances regarding the number of visited cells, as the obstacle ratio increases linearly from 10% to 40%. Figure 5a–g illustrate the number of visited cells by the five algorithms under different obstacle ratios.

The TPP and A* algorithms showed identical behavior, outperforming the BFS and Dijkstra algorithms, which also showed identical behavior across all trials. The GA presented the least performance and failed to find a path in large maps, as evident in Figure 5e–g, where the GA was intractable in maps larger than $64 \times 64$. Additionally, although increasing the obstacle ratio does not show a clear effect on the performance of the other algorithms, it negatively affected the GA's performance. In maps greater than $8 \times 8$ (Figure 5b–d), as the obstacle ratio increased, the GA could not find the shortest path. Moreover, even for small maps with a small number of obstacles, the GA visited significantly more cells than the other algorithms. Its behavior was inconsistent, i.e., the time required to find the path

varied greatly between the different trials due to the randomness involved in the design of this algorithm.

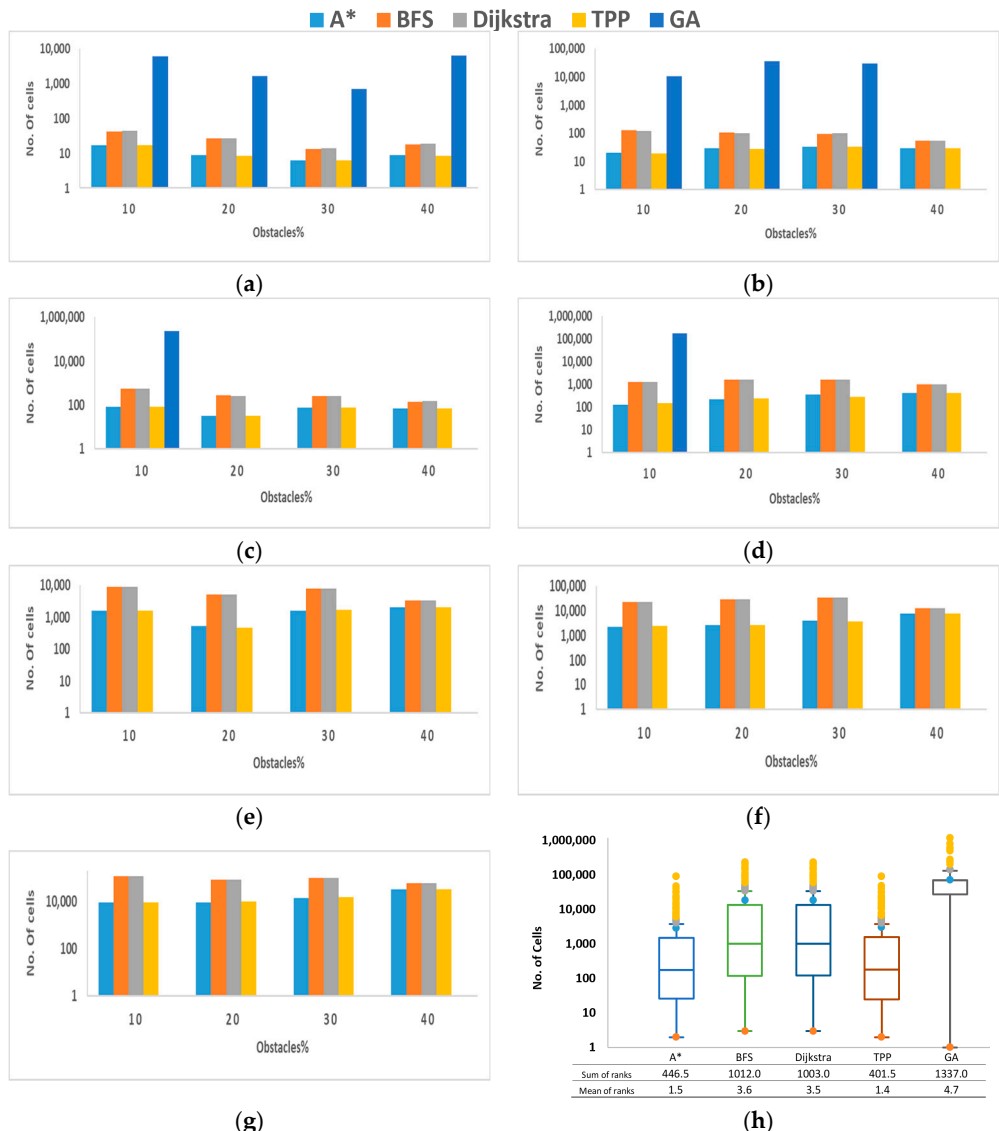

**Figure 5.** Robustness results: number of visited cells. (**a**) The average number of visited cells for 8 × 8 area. (**b**) The average number of visited cells for 16 × 16 area. (**c**) The average number of visited cells for 32 × 32 area. (**d**) The average number of visited cells for 64 × 64 area. (**e**) The average number of visited cells for 128 × 128 area. (**f**) The average number of visited cells for 256 × 256 area. (**g**) The average number of visited cells for 512 × 512 area. (**h**) Distribution of results for the number of visited cells.

To visually compare the distribution of the results of the robustness test, Figure 5h shows a boxplot. The plot confirms the scalability distribution findings (Figure 4b), where the TPP and A* algorithms have nearly similar distribution, as well as minimum and maximum values, which are better than the other algorithms. The BFS and Dijkstra algorithms result distributions were similar but significantly more dispersed and had higher and lower values than the TPP and A* algorithms. While the GA exhibited the highest minimum and maximum values with the least distribution of values as in all scenarios, the algorithm visited pointedly more cells to find a path or failed.

### 5.2. Path Length

5.2.1. Scalability Test

Comparing the resulting path length, illustrated in Figure 6a, it can be concluded that the TPP algorithm's discover path is similar to that of the BFS, Dijkstra, and A* algorithms, and is significantly shorter than that of the GA in all map sizes. As the map size increases, the length of the path found by the GA increases significantly and, as mentioned previously, the GA was intractable in large maps. Similar to the number of visited cells, increasing the map size increases the path length, which is also to be expected because more cells must be visited to reach more distant goals in larger maps.

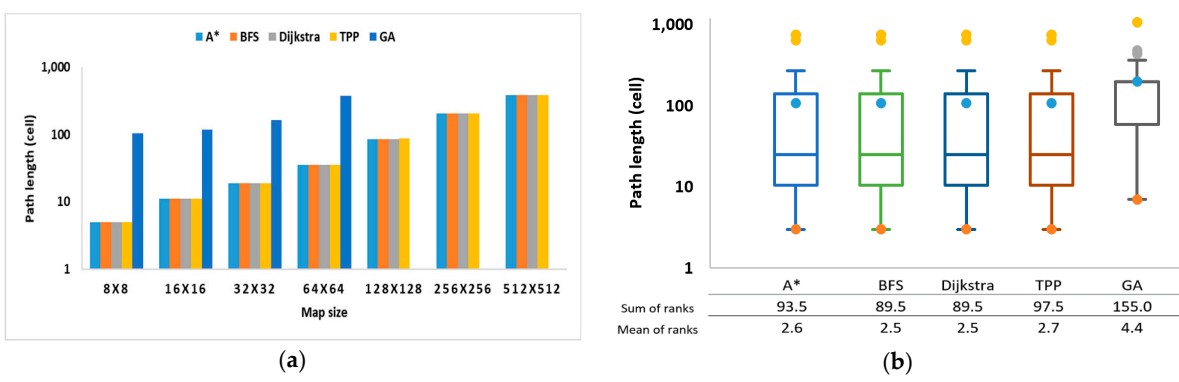

(**a**)  (**b**)

**Figure 6.** Scalability results: path length. (**a**) Average path length at different map sizes. (**b**) Distribution of results for path length.

When considering the results distribution represented in the boxplots in Figure 6b, the four algorithms—TPP, A*, BFS, and Dijkstra—produced paths of similar length. Moreover, their median lines are on the lower side of the box, indicating that most of the found path lengths were skewed to shorter values. On the other hand, the GA's discovered paths were much longer and, in large maps, the algorithm was intractable and found no path.

5.2.2. Robustness Test

As the obstacle ratio increases from 10 to 40%, the path length of each algorithm is illustrated in Figure 7a–g. The figure shows that the BFS, Dijkstra, A*, and TPP algorithms demonstrate identical path lengths in all runs, regardless of the obstacle ratio. However, the GA could not find the shortest path in large maps (i.e., it was intractable in maps larger than 8 × 8 with high obstacle ratios) (Figure 7b–d) and in maps larger than 64 × 64 (Figure 7e–g), even with lower obstacle ratios.

The result distributions of the path length robustness test for each algorithm are illustrated in the boxplots of Figure 7h. As shown in the figure, the TPP, A*, BFS, and Dijkstra algorithms have identical distributions and minimum and maximum values. On the other hand, the GA shows very low dispersion, as it was intractable in scenarios of large maps or high obstacle ratios. Additionally, its values are generally higher than the other algorithms.

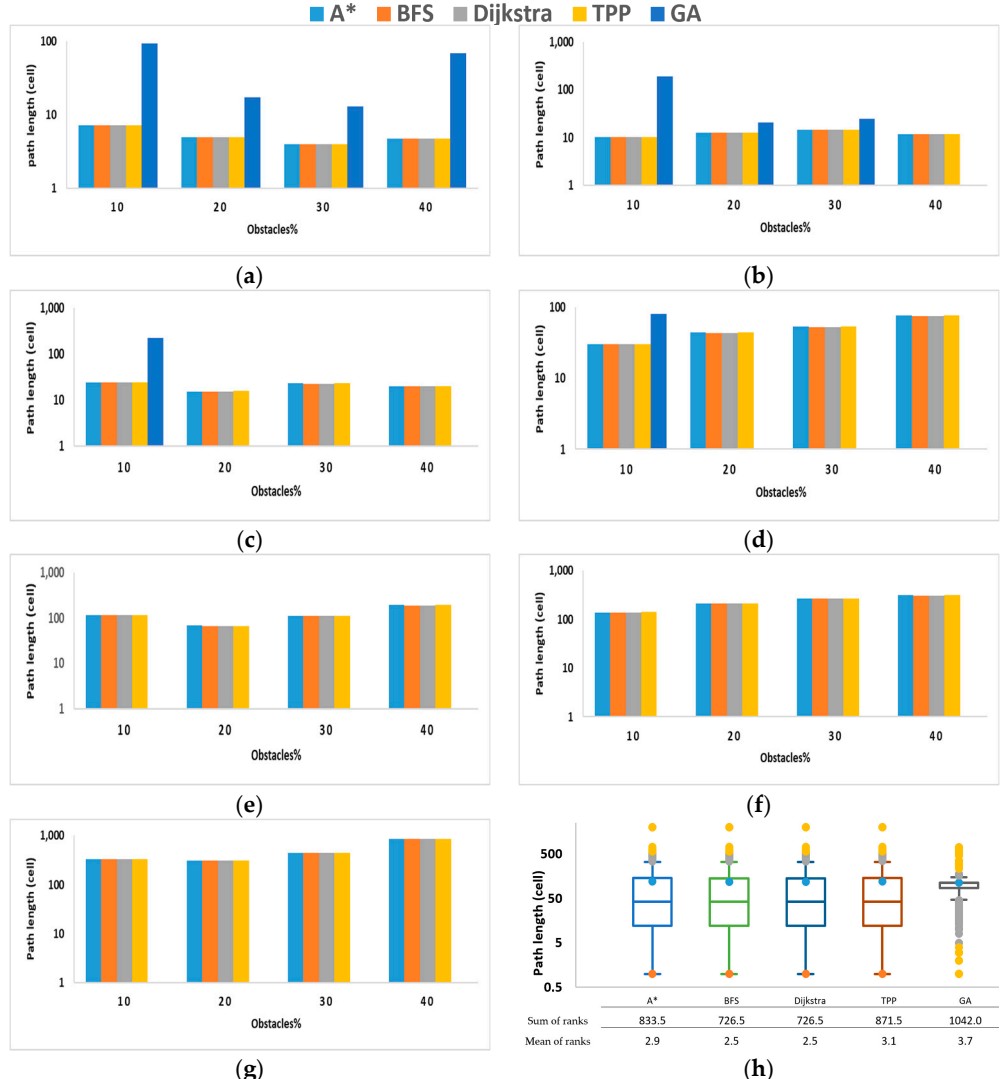

**Figure 7.** Robustness results: path length. (**a**) Average path length for 8 × 8 area. (**b**) Average path length for 16 × 16 area. (**c**) Average path length for 32 × 32 area. (**d**) Average path length for 64 × 64 area. (**e**) Average path length for 128 × 128 area. (**f**) Average path length for 256 × 256 area. (**g**) Average path length for 512 × 512 area. (**h**) Distribution of results for path length.

### 5.3. Execution Time

Based on the flowchart in Figure 3, the TPP control flow does not include any nested loops, and it has O(m × n) complexity, where m × n is the size of the map. Thus, the TPP algorithm has similar complexity to the A* algorithm. However, although some algorithms may have similar complexity when theoretically analyzed, they might differ in actual execution time. Therefore, in this paper, we have also evaluated each algorithm's exact execution time to compare their computational overheads. The execution time is measured as the real time from when the system calls an algorithm's main function till the control returns to the caller. It is important to note that all algorithms were run on the same machine under identical load and conditions for this measure to be meaningful.

#### 5.3.1. Scalability Test

As shown in Figure 8a, the TPP algorithm shows the fastest execution time (in milliseconds) compared with the other algorithms for all maps. The A* algorithm has the second-best performance. However, it is slower than the TPP algorithm in all maps. Next comes the BFS and Dijkstra algorithms, which have nearly similar results in large maps,

while in maps smaller than 64 × 64, the Dijkstra algorithm is relatively faster. The GA is significantly slower than the other algorithms and could not perform well in large maps; therefore, its line was discontinued as it did not find a path.

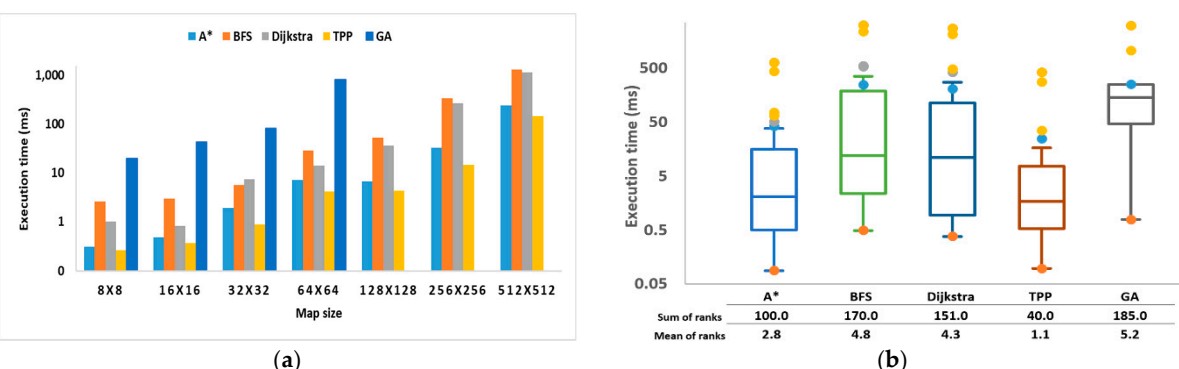

**Figure 8.** Scalability results: execution times. (**a**) Average execution time at different map sizes. (**b**) Distribution of results for execution time.

The execution time results that are shown in the boxplots in Figure 8b demonstrate that the TPP algorithm also produces better results distribution in terms of speed, compared with all benchmark algorithms, and the minimum and maximum execution times are lower than the other algorithms.

### 5.3.2. Robustness Test

As illustrated in Figure 9a–g, under different obstacle ratios in all area sizes, the TPP algorithm's execution time is lower than the benchmark algorithms, due to its beforehand obstacle avoidance strategy, which saves time to explore blocked paths. The A* algorithm shows marginal exceptions outperforming the TPP algorithm in a few scenarios. The performance gap between the TPP and A* algorithms on one side and the other algorithms, on the other side, increases as the map size increases at all obstacle ratios. In most scenarios, the Dijkstra and BFS algorithms show comparable execution times, which are significantly slower than the TPP algorithm. Akin to the previous performance measures, the GA exhibited the least performance, and it was not possible to report the GA results for scenarios with large maps or high obstacle ratios.

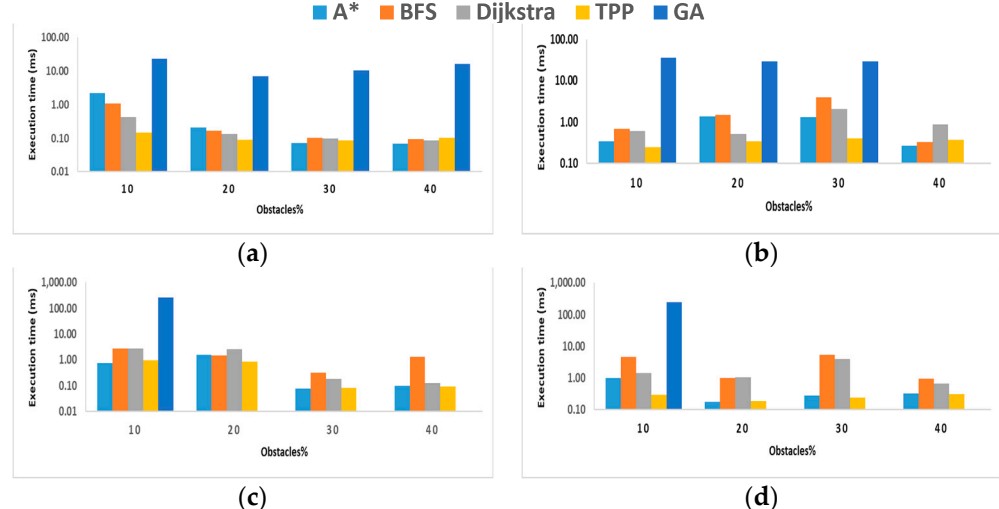

**Figure 9.** *Cont.*

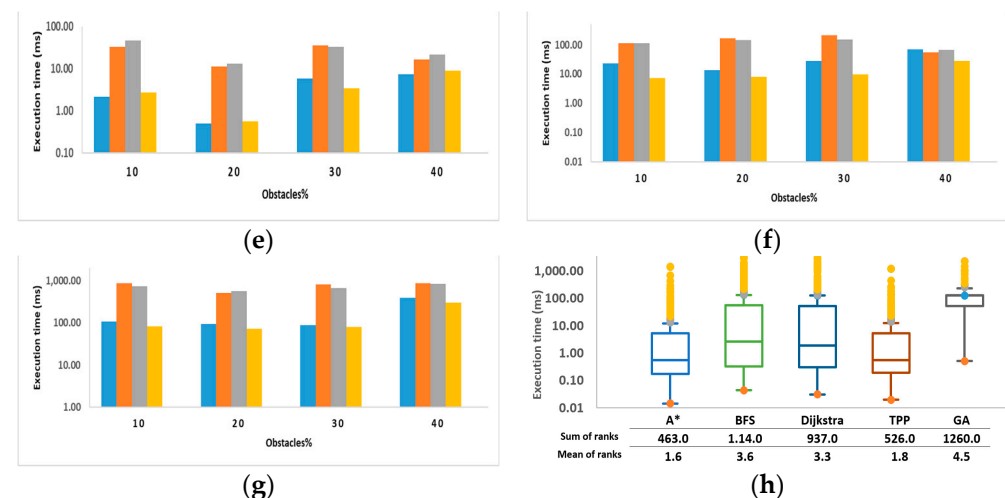

**Figure 9.** Robustness results: execution time in milliseconds. (**a**) Average execution time for 8 × 8 area. (**b**) Average execution time for 16 × 16 area. (**c**) Average execution time for 32 × 32 area. (**d**) Average execution time for 64 × 64 area. (**e**) Average execution time for 128 × 128 area. (**f**) Average execution time for 256 × 256 area. (**g**) Average execution time for 512 × 512 area. (**h**) Distribution of results for execution time.

The execution time result distributions are shown in the boxplots in Figure 9h. Similar to the number of visited cells, the TPP algorithm's execution time has a better statistical distribution, which is close to the A* algorithm but better than the BFS, Dijkstra, and GA. Furthermore, the TPP algorithm and GA show the lowest minimum and maximum values with positively skewed medians, indicating lower execution times. Again, as the GA was intractable in many scenarios and failed to find a path, its distribution is the worst and shows the highest minimum and maximum values.

To sum up, Table 1 presents the main findings of our experimental evaluation study, which indicates that, when scalability to large maps is considered, the proposed TPP algorithm outperformed all of the examined algorithms in terms of the number of visited cells and execution time. However, for the path length measure, all of the studied algorithms showed identical performance, where each was successful in identifying an optimal path except the GA, which halted in many scenarios before finding one. The same findings were reached when robustness to obstacles was the main focus, with the A* algorithm showing nearly identical performance to the TPP algorithm in terms of the number of visited cells. For the execution time, the TPP algorithm was faster in the majority of scenarios. The A* algorithm excelled only in small maps of large obstacle ratio.

**Table 1.** Summary of evaluation results, considering each design issue.

| Design Issue | Performance Measure | Best Algorithm | Scenario |
|---|---|---|---|
| Scalability | No. of visited cells | TPP | All |
| | Path length | TPP, A*, BFS and Dijkstra | All |
| | Execution time | TPP | All |

**Table 1.** *Cont.*

| Design Issue | Performance Measure | Best Algorithm | Scenario |
|---|---|---|---|
| | No. of visited cells | TPP and A* | All |
| | Path length | TPP, A*, BFS and Dijkstra | All |
| Robustness | Execution time | TPP | All scenarios except small maps ($\leq 8 \times 8$) of large obstacle ratios ($\leq 30\%$) |
| | | A* | small maps ($\leq 8 \times 8$) of large obstacle ratios ($\leq 30\%$) |

## 6. Conclusions

This paper introduced the TPP algorithm, a novel path planning algorithm, inspired by the natural tide phenomena. In this algorithm, the goal introduces an attraction force that pulls agents towards it, while obstacles produce a repulsive force that directs an agent away from them. Adopting the tide phenomenon allows TPP to reduce the number of paths it needs to search to find the optimal path.

Different experiments were conducted to measure the number of visited cells, path length, and algorithm speed. The proposed TPP algorithm performance was compared with four well-established path planning algorithms. The experimental results showed that the TPP algorithm finds a path that is equal to the path length found by the exact methods in significantly less time, while visiting fewer cells. Due to the centrifugal force property, the TPP algorithm could efficiently avoid obstacles in advance and reduce the time and energy required to discover an area. Thus, the proposed TPP algorithm is scalable to large maps and robust under high obstacle ratios. In the future, further tests on real robots will be conducted, in addition to testing the algorithm over more complex and dynamic environments. Sensor information and multi-sensor information fusion technologies will be considered to improve local path planning. Applying the algorithm to multi-robot missions, where task assignment and intercommunications between robots are important, will be researched.

**Author Contributions:** Conceptualization, H.K. and H.E.; formal analysis, S.A., H.E., H.Q. and B.A.; funding acquisition, H.K.; investigation, H.K., S.A., H.E., H.Q., B.A. and G.A.; methodology, H.K., S.A., H.E., H.Q., B.A. and L.A.-S.; project administration, H.K.; resources, H.K.; software, S.A., H.E., H.Q., B.A., L.A.-S. and G.A.; supervision, H.K.; validation, H.E., L.A.-S. and G.A.; visualization, H.E., L.A.-S. and G.A.; writing—original draft preparation, S.A., H.Q., B.A., L.A.-S. and G.A.; writing—review and editing, H.K. and H.E. All authors have read and agreed to the published version of the manuscript.

**Funding:** This research was funded by Researchers Supporting Unit, Project number (RSP- 2021/204), King Saud University, Riyadh, Saudi Arabia.

**Institutional Review Board Statement:** Not applicable.

**Informed Consent Statement:** Not applicable.

**Data Availability Statement:** Not applicable.

**Acknowledgments:** We would like to acknowledge what a wonderful student and friend Hajar Qahmash was. She has tragically passed away before she could see her name as a published author. She will be extremely missed.

**Conflicts of Interest:** The authors declare no conflict of interest.

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
