# Peer review of "Tide-Inspired Path Planning Algorithm for Autonomous Vehicles"

_remotesensing, doi:10.3390/rs13224644_

Round 1
Reviewer 1 Report
The authors have developed an interesting planetary-inspired approach to developing path planning based on the concept of tidal forces. This approach is shown to offer improvements over widely used alternatives.
The reviewer had trouble reading the various plots that were presented as summarizing performance. Since the entire point of the paper was to show the advantages of the novel tide approach, this should be corrected. One way would be to use much more obviously different symbols for each of the plot elements. Even an open circle for the TPP results would've been easier to identify.
Similarly, while the box plots are a useful qualitative way to show summary results for the various path planning method results, the reviewer--and the reader--will find themselves trying to extract numerical bounds. This would be solved if summary tables were also included.
By attempting to include results from a myriad of testing conditions, the fundamental gist of the effort becomes obscured and the number of plots degrades readability. For example, in Fig. 5, cases from 8x8 --> 512x512 are presented. The authors should consider selecting a subset of cases to clearly illustrate the general trend of the results. With fewer plots, they could be larger, easier to read, and help make the case for the method.
Fig. 7 has multiple plots that could all/mostly combined into a single plot. As already noted, the symbols are difficult to tease out of the plot. In fact, it really appears as if there are only one or two data series per plot, suggesting that they could all be presented in a single, legible plot. Further, since the axes are linear with identical spans, it would be easy--and informative--to have them on a single plot.
Similar comments for making Fig. 9 more readable.
The authors discuss their results on the way to their conclusions to highlight the key behaviors observed between the conventional and the TPP methods. The reader is left to form their own conclusions, but would be helped if the authors compactly summarized the the advantages/disadvantages of each. The reader will benefit from a compact summary that is easy to see and interpret. Users of path planning algorithms would be helped by such a summary: "Okay, for my problem, which is NxN, I see that I can get pretty good results using method Y." For researchers, presenting the performance matchups in a compact way can help inform how they present their results so it is easier to make apple-to-apple comparisons.
Amplification of future work would be useful. What are the obvious--or not so obvious--paths of inquiry to pursue?
A thorough read of the manuscript will help reveal some grammar tweaks that will improve the readability. Examples:
Introduction. "...accuracy in one sides and time and energy in the other side..." Consider: "...accuracy on one hand and time and energy on the other..."
"...test optimality), and algorithm..."
"...shortest path, if it exists, at significantly..."
Literature Review.
Not all acronyms are defined. While the typical reader may be versed in this area, efforts to help new readers are important. Therefore, define BFS, etc.
Typically, do not begin sentences with a reference number--e.g., "[10] introduced..." and others.
Algorithm Design
"...then the so far visited cells from..." Consider, "...then the cells visited from..." or "...then the sequence of cells visited from..."
Evaluation methodology
- "The number of visited cells, which..."
Reviewer 2 Report
This paper proposed a Tide Path Planning (TPP) algorithm for autonomous vehicles, which is inspired by the tide phenomenon in nature and belongs to heuristic algorithms. In the TPP algorithm, an agent is pulled to the goal while the obstacles direct the agent away from the goal, which basic idea is simple. The proposed algorithms were compared with A*, BFS, Dijkstra and GA, in which the number of visited cells, path length and algorithm execution time were chosen as the measures to indicting the energy efficiency, testing optimality, and access the algorithm complexity, respectively.
The algorithm should be re-designed and academic writing should be improved.
Critical issues:
- Is the TPP algorithm deals with static path planning only? Could the TPP algorithm deals with the heterogeneous network? What is the difference between the proposed TPP algorithm with the gravity model in trip assignment?
- The proposed algorithm is simple, which is path planning with weighted cells.
line 128, ‘…local exploration processes and execution time. Therefore, this paper aims to help in bridging this gap.’ How the proposed TPP bridging this gap between local exploration process and execution time? As from Figure 4 and Figure 5, the scalability and robustness of the number of visited cells and robustness tests show no significant improvement to A*. For the execution time, as the unit is millisecond, the improvement is little.
- Why the authors compare the proposed TPP algorithm with A*, BFS and so on, but not with other nature-inspired heuristics (such as what the authors have mentioned PGPP, GSA)?
Minor issues:
- line 31, ‘a source point to a goal point’, in the transportation area, that is called origin and destination.
- line 36, the classification of path planning algorithms is incomplete. On the same note, the literature review section should be improved.
- line 42, ‘However, they cannot guarantee finding the optimal solution.’ What optimal?
- line 82, ‘A* is both complete and optimal,…’ On what condition, A* is complete and optimal? Additionally, when you review path planning related algorithms, besides complete and optimal and time, space should be discussed too.
- line 180: How this work define the map for path planning? Could you describe what the network is? Homogeneous or heterogeneous?
- line 236: obstacle ratio increases linearly from 5% to 40%.
- Figure 5 is also the scalability results?
Round 2
Reviewer 2 Report
The authors have roughly addressed the reviewer's comments. The figure quality should be improved expecially for Fig. 4-9. Basically, the authors should employ vectorgraph and make the font same as main text.
Author Response
Thankfully acknowledging the suggestion to improve the quality of the graphs by employing vectorgraph and making the font same as main text. However, it seems like due to the many changes on the document using track changes, some graphs are not displaying correctly. Accordingly, we have had to place the new graphs in a separate file, instead of altering the manuscript, to ease the reviewing and editing processes.
